# Mosquitoes Possess Specialized Cuticular Proteins That Are Evolutionarily Related to the Elastic Protein Resilin

**DOI:** 10.3390/insects14120941

**Published:** 2023-12-11

**Authors:** Sakura Ohkubo, Tohki Shintaku, Shotaro Mine, Daisuke S. Yamamoto, Toru Togawa

**Affiliations:** 1Department of Biosciences, College of Humanities and Sciences, Nihon University, Sakurajyosui 3-25-40, Setagaya-ku, Tokyo 156-8550, Japanmines013@affrc.go.jp (S.M.); 2Division of Insect Advanced Technology, Institute of Agrobiological Sciences, National Agriculture and Food Research Organization, Owashi 1-2, Tsukuba 305-8634, Japan; 3Division of Medical Zoology, Department of Infection and Immunity, Jichi Medical University, Yakushiji 3311-1, Shimotsuke 329-0498, Japan; daisukey@jichi.ac.jp

**Keywords:** resilin, resilin-related, cuticular protein, mosquito

## Abstract

**Simple Summary:**

Insects possess elastic cuticles in movable body parts such as the wing hinges and the basement of the hind legs of jumping insects. These cuticles contain a rubber-like protein called resilin. Resilin is a member of a cuticular protein family with a specific binding domain to chitin, which is another component of the cuticle. Resilin, or resilin-like proteins, have been reported in many insects. In a malaria vector mosquito, a curious cuticular protein has been found. It has a chitin-binding domain that is highly homologous to resilin, but it lacks repetitive sequences that are common in resilin and are essential for its elastic properties. It was unknown whether this protein is conserved in mosquitoes, whether this protein is the mosquito resilin, or whether mosquitoes possess another canonical resilin. In this paper, we found that this protein (termed resilin-related) is conserved in mosquitoes and seems to be evolutionarily derived from the resilin. Resilin-related, however, has very different structural features from resilin. Therefore, it probably plays special roles distinct from those of resilin in mosquitoes. Another resilin-like protein was found to exist commonly in holometabolous insects. Although this protein is likewise not a canonical resilin, it might confer elastic properties to cuticles in mosquitoes.

**Abstract:**

Resilin is an elastic protein that is vital to insects’ vigorous movement. Canonical resilin proteins possess the R&R Consensus, a chitin-binding domain conserved in a family of cuticular proteins, and highly repetitive sequences conferring elastic properties. In the malaria vector mosquito, *Anopheles gambiae*, however, a cuticular protein has been found that has an R&R Consensus resembling that of resilin but lacks the repetitive sequences (here, we call it resilin-related or resilin-r). The relationship between resilin-r and resilin was unclear. It was also unknown whether resilin-r is conserved in mosquitoes. In this paper, phylogenetic and structural analyses were performed to reveal the relationship of resilin homologous proteins from holometabolous insects. Their chitin-binding abilities were also assessed. A resilin-r was found in each mosquito species, and these proteins constitute a clade with resilin from other insects based on the R&R Consensus sequences, indicating an evolutionary relationship between resilin-r and resilin. The resilin-r showed chitin-binding activity as same as resilin, but had distinct structural features from resilin, suggesting that it plays specialized roles in the mosquito cuticle. Another resilin-like protein was found to exist in each holometabolous insect that possesses resilin-like repetitive sequences but lacks the R&R Consensus. These results suggest that similar evolutionary events occurred to create resilin-r and resilin-like proteins.

## 1. Introduction

The insect body is covered by cuticles, which function as the exoskeleton and have a variety of mechanical properties. The cuticle is mainly composed of the polysaccharide chitin and structural cuticular proteins [1,2,3]. The variations in the mechanical properties of the cuticle are thought to depend on the degree of sclerotization, i.e., cross-linking between proteins and between proteins and chitin [4,5] and the variation in composing cuticular proteins [1,2,3]. Insect cuticular proteins have been classified into several families. Most of the proteins, however, are members of the CPR family, named after their conserved R&R Consensus, which functions as a chitin-binding domain [2,3,6,7].

Resilin is one of the CPR members found in elastic rubber-like cuticles [8,9,10]. A precursor of resilin (pro-resilin) is secreted from the epidermal cells into the subcuticular space and then cross-liked to form a three-dimensional, easily deformable protein network of functional resilin [11]. Resilin provides elastic properties in many movable parts, such as wing hinges, the basements of the hind legs of jumping insects, and the tendons of flight muscle [8,12], playing important roles for insect behavior, such as kinetic energy storage and muscle shock absorbers [13]. The elastic property of resilin is considered to be derived from highly repetitive sequences in the regions flanking the R&R Consensus [10,14,15]. Resilin or resilin-like proteins have been reported or annotated in a variety of insect species [14,16]. A malaria vector mosquito, *Anopheles gambiae*, has a peculiar CPR, called Agam_CPR152, related to resilin [17]. Although the R&R Consensus of this protein highly resembles that of resilin from other insects, the flanking region is quite different. The relationship between this protein and the resilin was unclear, and it was unknown whether this protein is conserved among mosquitoes. In this study, we found orthologous proteins of Agam_CPR152 in several other mosquito species whose R&R Consensus is homologous to that of resilin. The flanking regions of these proteins, however, lacked repetitive sequences and showed characteristically different amino acid compositions compared to resilin from other insects, suggesting unique functions in mosquitoes.

## 2. Materials and Methods

### 2.1. Insects

The *Anopheles stephensi* strain SDA500 was maintained for multiple generations at Jichi Medical University, Japan. The *Bombyx mori* strain p50 was provided by the National Bio-Resource Project (NBRP) of Japan. The *Drosophila melanogaster* strain Oregon-R was kindly provided by Dr. Keisuke Kamimura at the Tokyo Metropolitan Institute of Medical Science (TMIMS, Tokyo, Japan). The wild-type strain of *Tribolium castaneum* and a mutant strain *yellow fat body* of *Athalia rosae* were provided by the National Agriculture and Food Research Organization (NARO, Tsukuba, Japan) and had been maintained for several generations at Nihon University, Japan.

### 2.2. Sequence Analyses

The region of the R&R Consensus of CPR proteins was defined according to Cornman et al. (2008) [17] and Willis (2010) [18], i.e., beginning with two amino acid residues N-terminal to the “aromatic triad” and ending with nine residues C-terminal to the final invariant glycine. For the phylogenetic analysis, sequences were aligned by ClustalW (Appendix A), and a neighbor-joining tree was constructed by MEGA11 [19]. Signal peptides were predicted by SignalP 6.0 (https://services.healthtech.dtu.dk/service.php?SignalP) (accessed from 1 March 2023 through 30 November 2023) [20]. Analyses of amino acid composition and sequences were performed using MacVector 18.5 (MacVector, Apex, NC, USA). Principal component analysis (PCA) was performed to assess the variance of amino acid composition using the R software program (version 4.0.5) [21]. The proportion of amino acid residues in each protein was used for PCA. Repetitive sequences were compared by WebLogo 3 (https://weblogo.threeplusone.com) (accessed on 2 December 2023) [22,23].

### 2.3. cDNA Cloning

Total RNA samples were isolated from *Anopheles stephensi* (whole bodies of male pharate adults at 36 h after pupation), *D. melanogaster* (body trunks without heads of male adults just after eclosion), *T. castaneum* (whole bodies of pupae at various developmental stages), *B. mori* (forewings of male pharate adults 7 days after pupation), and *Athalia rosae* (whole bodies of female pharate adults at 1 day before eclosion) using RNAiso Plus (Takara Bio, Kusatsu, Japan). First-strand cDNA was synthesized from 1 μg of total RNA using an oligo(dT)20 primer (Toyobo, Osaka, Japan) and ReverTra Ace reverse-transcriptase (Toyobo) in a 20 μL reaction mixture. The PCR primers for the amplification of target cDNA sequences were designed based on the sequences registered in GenBank/Embl/DDBJ (Appendix A). RT-PCR was performed using KOD Plus Neo (Toyobo) with the primers listed in Appendix A and the first-strand cDNA described above as a template. The RT-PCR products were cloned into pUC19 using In-Fusion Snap Assembly Master Mix (Takara Bio) or pGEM-T Easy (Promega, Madison, WI, USA) after the addition of dA. The nucleotide sequences of the cDNA clones were determined by Sanger sequencing using the BigDye Terminator v3.1 Cycle Sequencing Kit (Thermo Fisher Scientific, Waltham, MA, USA) and 3500 Genetic Analyzer (Thermo Fisher Scientific). The cloned cDNA sequences were registered in the GenBank/EMBL/DDBJ databases (Appendix A).

### 2.4. Construction of Expression Plasmids and Preparation of Recombinant Proteins

To insert His-tag into the expression vector pGEX-6P-1 (Cytiva, Tokyo, Japan), 6xHis_NtF1 and 6xHis_NtR1 (Appendix A) were annealed in a solution containing 20 μM each of oligonucleotide, 10 mM of Tris-HCl, 100 mM of NaCl, and pH 7.5 by heating at 95 °C for 10 min, followed by a stepwise decrease of 5 °C with a 5 min hold at each temperature until 25 °C, and then inserted into the *Not*I site of pGEX-6P-1. The resulting expression vector was named pGEX-6P-1-NtHis. The R&R Consensus regions of *Dmel_resilin*, *Bmor_CPR140*, and *Aste_resilin-r* were amplified by PCR using their cDNA plasmid clones as templates and *Eco*RI/*Xho*I-tagged gene-specific primers (Appendix A). The amplified fragments were digested by *Eco*RI and *Xho*I and inserted between those sites of pGEX-6P-1-NtHis. The resultant expression plasmids were named pGEX6P-DmelResilinRR-His, pGEX6P-BmorCPR140RR-His, and pGEX6P-AsteResilin-rRR-His, respectively. The sequences of all inserts in the expression plasmids were confirmed by Sanger sequencing.

Expression plasmids were expressed in the *Escherichia coli* Rosetta 2 strain (Novagen, Darmstadt, Germany). Bacteria with each construct were grown in 300 mL of LB with 50 μg/mL ampicillin and 34 μg/mL chloramphenicol at 30 °C. When the optical density at 600 nm reached 0.5, IPTG was added to a final concentration of 0.5 mM. After incubation for a further 4 h, bacterial cells were collected by centrifugation and resuspended in 30 mL of PBS (137 mM NaCl, 8.1 mM Na_2_HPO_4_, 2.68 mM KCl, 1.47 mM KH_2_PO_4_, pH 7.4). The cell suspensions were stored at −80 °C until use. After the bacterial cells in the suspension were disrupted by sonication, the insoluble fraction was removed by centrifugation. The recombinant proteins in the soluble fraction were purified with Glutathione Sepharose 4 Fast Flow (Cytiva), equilibrated with PBS, and eluted with 10 mM reduced glutathione-containing buffer. After buffer exchange by dialysis, the recombinant proteins were further purified with Ni Sepharose 6 Fast Flow (Cytiva) equilibrated with binding buffer (20 mM Na-phosphate buffer, pH 7.4, 500 mM NaCl, and 40 mM imidazole) and eluted with 500 mM imidazole-containing buffer according to the manufacturer’s instructions. The purified recombinant proteins were dialyzed against the chitin-binding buffer (see below). The homogeneity of the purified proteins was confirmed by SDS-PAGE, followed by Coomassie Brilliant Blue staining (Appendix A).

### 2.5. Chitin-Binding Assay

Chitin-binding activity was examined by chitin-affinity chromatography, as described in a previous study [7], with some modifications. Powdered chitin (from shrimp shells; Sigma-Aldrich, Darmstadt, Germany) was washed well with 0.1 N HCl and then 0.1 N NaOH, rinsed with distilled water, and suspended in the chitin-binding buffer (20 mM Na-phosphate buffer, pH 6.4, 150 mM NaCl, 0.1% Triton X-100). Each 200 μg of purified recombinant protein in 200 μL of chitin-binding buffer (1 mg/mL) was loaded on a 200 μL chitin column equilibrated with chitin-binding buffer. The flow-through was loaded onto the same column four more times. The column was washed 3 times with 1 mL of the chitin-binding buffer. The bound proteins were eluted with 200 μL of 8 M urea in binding buffer 7 times with 1 min intervals. The proteins in each fraction were analyzed by SDS-PAGE (Appendix A) and quantified by densitometry. The chitin-binding activity was calculated as a ratio of the amount of proteins eluted from the chitin column to that of proteins applied to the column.

## 3. Results

### 3.1. Resilin Homologous Proteins in Holometabolous Insects

Although resilin was identified in hemimetabolous insects at first, it was also identified and characterized in *Drosophila melanogaster* [10,11,24]. Therefore, the distribution and phylogenetic relationship of resilin homologous proteins were examined in holometabolous insects, including mosquitoes. For this purpose, CPRs homologous to resilin were searched by tblastn in four mosquito species (*Anopheles gambiae*, *Anopheles stephensi*, *Aedes aegypti*, and *Culex pipiens pallens*), sand fly (*Lutzomyia longipalpis*), fruit fly (*D. melanogaster*), tsetse fly (*Glossina morsitans*), red flour beetle (*Tribolium castaneum*), Asian long-horned beetle (*Anoplophora glabripennis*), silkworm (*Bombyx mori*), fall armyworm (*Spodoptera frugiperda*), turnip sawfly (*Athalia rosae*), and a parasitoid wasp (*Nasonia vitripennis*) using the pro-resilin of *D. melanogaster* (Dmel_Resilin, NM_137313) as a query against the nonredundant (nr) sequence collection of GenBank/EMBL/DDBJ or transcript collection of VectorBase (https://vectorbase.org/vectorbase/app) (accessed from 23 August 2022 through 24 November 2023). We found two homologous sequences from each species (Figure 1). Because most of them were predicted mRNA sequences, the cDNA for the most similar sequences to the pro-resilin from *Anopheles stephensi*, *D. melanogaster*, *T. castaneum*, *B. mori*, and *Athalia rosae* was cloned to confirm the primary structure of the proteins. For the second most similar sequence in *B. mori* (Bmor_CPR139), another 3′ structure was predicted in the KAIKObase (https://kaikobase.dna.affrc.go.jp) (accessed on 12 February 2021). Therefore, its 3′ sequence was also cloned. The cloned sequences were registered in the GenBank/EMBL/DDBJ databases (Appendix A). The phylogenetic relationship of their proteins was analyzed based on the R&R Consensus (Figure 1). The third and fourth most similar proteins to Dmel_Resilin in *D. melanogaster*, *Anopheles gambiae*, and *T. castaneum* were used as an outgroup. Mosquito proteins found in each species constituted one clear clade (tentatively called Group I in this study). The proteins from other species that were most similar to Dmel_Resilin in these species, tentatively called Group II, constituted a clade with Group I proteins, although the support of this clade by bootstrap value was not so high. Group II included Dmel_Resilin and *Tribolium* pro-resilin (Tcas_CPR103) [25]. The secondary proteins found by tblastn using Dmel_Resilin as a query (Group III) were isolated from the outgroup with Groups I and II (Figure 1). Group III includes *D. melanogaster* Cpr56F, which was the second candidate of *Drosophila* resilin with sequences significantly similar to tryptic peptides of an authentic resilin from locust, *Schistocerca gregaria* [11]. This result suggests that these resilin homologous proteins in Groups I, II, and III are derived from an ancestral resilin protein in Holometabola.

### 3.2. Structure of Resilin Homologous Proteins

Andersen (2010) [14] proposed the following definitive features of the primary structure of resilin: (1) the R&R Consensus, which functions as a chitin-binding domain and is the defining feature of the CPR family of cuticular proteins; (2) the signal peptide, conferring the property of secretion into the extracellular matrix, such as the cuticle; and (3) a series of short repeats of amino acid residues containing glycine and proline, conferring long-range elasticity after the maturation of resilin proteins, including cross-linking. Therefore, we examined the resilin homologs for these features. Because all resilin homologs described above are CPRs, they have R&R Consensus and signal peptides (Figure 2 and Appendix A). Group II proteins from *D. melanogaster* (Dmel_Resilin) and *T. castaneum* (Tcas_CPR103), which have been recognized as pro-resilin, possessed repetitive sequences at both the N-terminal and C-terminal to the R&R Consensus (Figure 2, Appendix A), as reported previously [14]. All other Group II proteins also possessed two types of repetitive sequences at N-terminal and C-terminal region, respectively, except for *Anoplophora glabripennis* LOC108907992, which had a single type of repeat at the N-terminal region to the R&R Consensus. In contrast, Group I proteins from mosquitoes showed no repetitive sequences (Figure 2). Many Group III proteins do not have repetitive sequences, but some of them showed either N-terminal or C-terminal repetitive sequences (Figure 2 and Appendix A). Resilin is also known to have a conserved N-terminal sequence (xxEPPVNSYLPPS) [14]. All Group II proteins contained this sequence (Figure 2, Appendix A). This sequence was also found in Group I proteins. Group III proteins possessed divergent versions of the N-terminal sequences (Figure 2 and Appendix A).

We have noticed that *Anopheles gambiae* Group I protein (Agam_CPR152) contains a large quantity of histidine residues [17], which is not observed in pro-resilin [14,18]. To determine whether this feature is common among Group I proteins, the amino acid composition of Group I proteins was analyzed and compared with Group II and III proteins (Appendix A). Group II proteins contain many glycine, serine, and proline residues (Appendix A), which is one of the characteristic features of resilin [14]. The composition of Group III proteins was more variable, with a tendency to contain many glycine, tyrosine, and proline, or asparagine. The most remarkable feature of Group I proteins was the high proportion of histidine residues (Appendix A). Principal component analysis of the proportions of each amino acid residue also showed that Group I proteins have distinct amino acid contents from Group II and III, and histidine is a characteristic residue of Group I proteins (Figure 3). These results suggest that Group II proteins possess all the definitive features of resilin and probably function as resilin, conferring elastic properties on the cuticle in each species. On the other hand, although Group I proteins, which are specific to mosquitoes, possess an R&R Consensus highly homologous to that of resilin, they might provide another property to the cuticle. Because these proteins are evolutionary related to resilin, but would play different roles, we call mosquito-specific Group I proteins resilin-related (resilin-r).

### 3.3. Chitin-Binding Activity of Resilin and Resilin-Related

The R&R Consensus functions as the chitin-binding domain of CPRs [6,7], and *Drosophila* resilin has been shown to bind to chitin via the R&R Consensus [26]. Therefore, to determine whether mosquito resilin-r has the same ability to bind to chitin via the R&R Consensus as other resilin proteins, the chitin-binding activities of recombinant proteins with the R&R Consensus region of resilin/resilin-r were analyzed (Figure 4). In these recombinant proteins, the R&R Consensus was flanked by Glutathione S-transferase (GST) at the N-terminal and His-tag at the C-terminal. Recombinant protein without the R&R Consensus (GST-His) was also analyzed as a negative control. A recombinant protein with the R&R Consensus of *Anopheles stephensi* resilin-r (Aste_Resilin-r) showed chitin-binding activity as similar as that of resilin from *D. melanogaster* (Dmel_Resilin) and *B. mori* (Bmor_CPR140), although the control protein showed a trace level of binding (Figure 4). This result indicates that mosquito resilin-r proteins bind to chitin in the cuticle via their R&R Consensus.

### 3.4. Other Cuticular Proteins with Repetitive Sequences

The results described above suggested that mosquito resilin-r proteins with an R&R Consensus most homologous to that of resilin might play a different role from resilin in the cuticle and that mosquitoes do not possess canonical resilin proteins. Although mosquitoes possess Group III proteins, they have no repetitive region that is important for elastic properties (Figure 2). Which protein plays the role of resilin in mosquitoes? We noticed that the tblastn search using Dmel_Resilin as a query described in Section 3.1 found another secretory protein without the R&R Consensus from each mosquito species (tentatively called Group IV). Similar proteins were also found in other species (Appendix A), and these proteins have been mentioned as resilin-like proteins [3,14]. Moreover, peptides of *Anopheles gambiae* Group IV protein, AGAP002367, were detected in adult cuticle preparation [27], indicating that they are authentic cuticular proteins. They shared the conserved N-terminal sequences with resilin homologous proteins (Appendix A). Although PCA showed the tendency of Group IV proteins to have more serine, proline, and alanine residues than Group II proteins (Figure 3), the majority of amino acid residues are the same as those of Group II (resilin) proteins (Appendix A). Moreover, Group IV proteins possessed highly repetitive sequences as well as resilin proteins (Figure 2 and Appendix A), suggesting that these proteins might have elastic properties.

## 4. Discussion

Resilin or resilin-like proteins have been reported from many insect species [9,11,14,28]. Some predicted gene products are annotated as pro-resilin or pro-resilin-like in the GenBank/Embl/DDBJ nr database (Appendix A). But their structural and evolutionary relationship is confused. Here, we showed that a single resilin orthologous sequence is found in each holometabolous species based on the R&R Consensus (Groups I and II in Figure 1). All these resilin “orthologs” except mosquito proteins possess all sequence features of resilin, such as repetitive sequences and the conserved N-terminal sequence [14]. The repetitive sequences are considered to be essential for the elastic property of resilin [10]. The repetitive sequences shared core sequences at the N-terminal and C-terminal regions of the R&R Consensus, respectively (Figure 5). These sequences are consistent with those of Dmel_Resilin described by Ardell and Andersen (2001) [11]. They noticed that the second proline of the N-terminal repeat and the single proline of the C-terminal repeat, which corresponds to the second proline in Figure 5B, are typically followed by two glycine residues. Those short sequences containing proline and glycine have been found in other elastic structural proteins such as elastin and spider flagelliform silk, which are suggested to form β-turns and β-spiral [11]. The current study showed that the first proline followed by two serine residues is more conserved than the second proline followed by two glycine residues in the N-terminal region (Figure 5A). Because serine is also a small residue, as is glycine, this sequence might contribute to the formation of β-turn. Therefore, Group II proteins seem to confer elastic properties to the cuticle as authentic resilin. Group III proteins are more obscure. Although they possess divergent versions of the conserved N-terminal sequence and comparable amino acid compositions to Group II proteins, only some of them have repetitive sequences in either the N-terminal or C-terminal regions (Appendix A). These sequences somewhat match the repeat sequence of Group II proteins (Figure 5A,B). Group III proteins, if not all, might perform some resilin-like function. On the other hand, mosquito-specific Group I proteins, which we call resilin-related or resilin-r, showed neither repetitive sequences nor amino acid compositions similar to other resilin homologous proteins, although they contain the R&R Consensus highly homologous to that of resilin and the conserved N-terminal sequence (Figure 2 and Figure 3). Therefore, resilin-r seems unlikely to play the role of resilin.

Although *resilin-r* genes are conserved in mosquito species, this gene was not found in other holometabolous insects, even in *L. longipalpis*, another species of Nematocera (Figure 1). Mosquitoes seem to possess resilin-r (Group I) instead of canonical resilin (Group II). Therefore, an evolutionary event might have occurred in the *resilin* gene in the mosquito lineage after divergence from other groups of Diptera. Almost the entire region of the R&R Consensus of resilin/resilin-r is encoded by a single exon in many species (Figure 2). Therefore, the exon encoding the R&R Consensus in the *resilin* gene in the early ancestor of mosquitoes might have transferred into another locus to form the *resilin-r* gene by an exon shuffling mechanism [29]. Retroposition would be another possibility [30]. Although alternative splicing is hardly detected in CPR mRNAs, an isoform lacking the R&R Consensus is known in *Dmel_resilin* (FlyBase, https://flybase.org) (accessed on 17 January 2019). We also found a similar isoform from *B. mori* (*Bmor_CPR140* isoform B, accession no. LC782963). These facts might be related to the possibility of retoroposition to create the *resilin-r* gene. Group IV proteins lack the R&R Consensus but possess repetitive sequences and the conserved N-terminal sequence (Figure 2). Interestingly, the repetitive sequences of Group IV proteins shared the core sequences with the N-terminal repetitive sequences of Group II proteins (Figure 5). The consensus sequence seems to be PSS[S/Q]YGAP. This conservation suggests an evolutionary relationship between Group II and Group IV proteins. The locus or transcripts left behind after the removal of the R&R Consensus from the *resilin* gene might have become the gene for Group IV protein by a similar mechanism to create the *resilin-r* gene in mosquito. The timing of these evolutionary events, however, must be different because the *resilin-r* gene is specific to mosquitoes but the genes for Group IV protein are found in other species. More studies are necessary to elucidate the evolution of the *resilin* gene in mosquitoes and other insects.

Resilin plays an important role in conferring elastic properties to the specific movable parts of insects [8,12]. Recently, it was reported that *Dmel_resilin* is expressed at many movable joints in adult flies [24]. On the other hand, we have reported that *Agam_CPR152* (Group I, *resilin-r*) is expressed specifically in male pharate adults [31]. This data suggests that resilin-r probably plays a male-specific role in adult mosquitoes. Indeed, Agam_CPR152 was detected in male Johnston’s Organ by proteomics [27] and immunostaining [32]. Instead of resilin-r, Group IV proteins might confer elastic properties to the cuticle in mosquitoes even though they lack the chitin-binding R&R Consensus. The recombinant Dmel_Resilin protein shows elastic properties without the R&R Consensus [33]. Moreover, the recombinant Group IV protein of *Anopheles gambiae* was synthesized and showed elastic properties [34,35]. As described above, repetitive sequences of Group IV proteins contain the consensus core sequence, PSS[S/Q]YGAP, shared with repetitive sequences of canonical resilin (Group II). Although the R&R Consensus of resilin was shown to be important for its function in the cuticle [24], Group IV proteins might bind to chitin via other CPR proteins.

## 5. Conclusions

Although many resilin or resilin-like sequences have been reported or annotated, their relationship is confused. In this study, we clarified their relationship in holometabolous insects, especially in mosquitoes. Holometabolous insects appear to possess a single canonical resilin (Group II) in each species, conferring elastic properties to the cuticle. They also appear to possess a single secondary resilin homolog (Group III), although their function and phylogenetic divergence from Group II are not clear. Other resilin-like protein (Group IV) are likely common in holometabolous insects. These proteins lack the chitin-binding domain but contain repetitive sequences that share the consensus sequence with the canonical resilin (Group II), suggesting the contribution of elasticity to the cuticle. In the mosquitoes, the canonical *resilin* gene (Group II) might have been lost and replaced by the *resilin-r* gene (Group I). Resilin-r may play a mosquito-specific role, and instead Group IV protein may play a resilin-like role in mosquito cuticle, although their functions should be elucidated in future studies.

## Figures and Tables

**Figure 1 insects-14-00941-f001:**
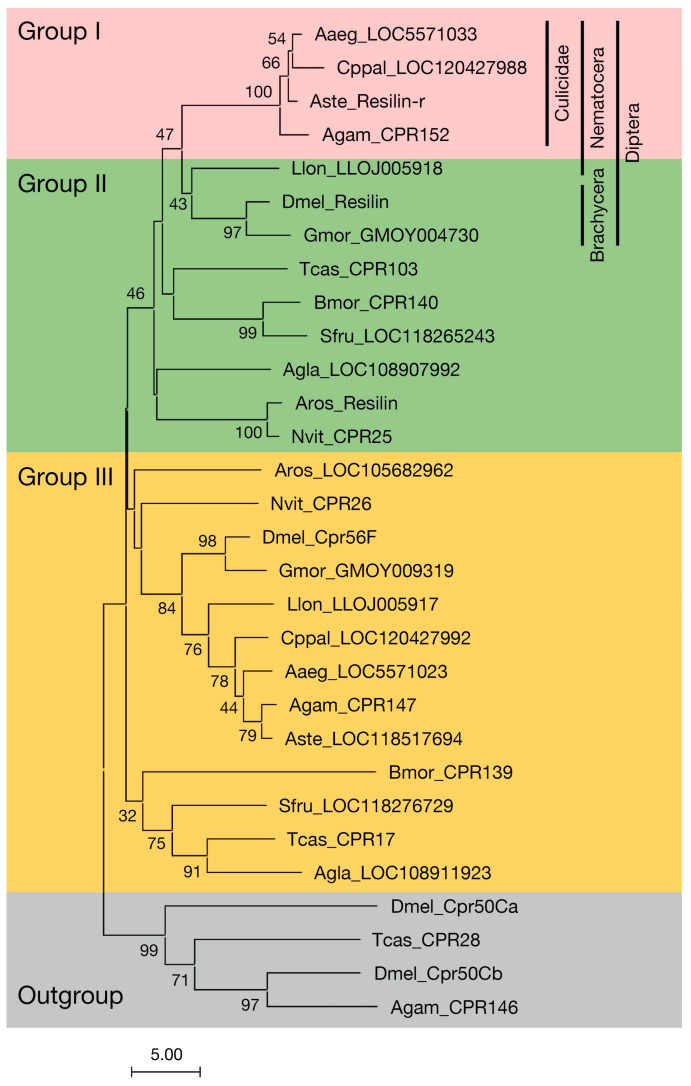
Phylogenetic relationship between CPR proteins homologous to resilin based on the R&R Consensus. The amino acid sequences of R&R Consensus were aligned by ClustalW, and a neighbor-joining tree was constructed by MEGA11. The branch lengths represent the evolutionary differences calculated using the number of differences. The bootstrap values (1000 replicates) that are higher than 30 are shown at the branch points. *D. melanogaster* Cpr50a, Cpr50b, *Anopheles gambiae* CPR146, and *T. castaneum* CPR28, which are the 3rd and 4th most similar to resilin in these species, were used as an outgroup. The tentative grouping in this study is shown by different color background (Group I, red; Group II, green; Group III, orange). The taxonomy of dipteran sequences in Group I/II is indicated to the left. Letters leading the sequence names represent insect species: Aaeg, *Aedes aegypti*; Agam, *Anopheles gambiae*; Agla, *Anoplophora glabripennis*; Aste, *Anopheles stephensi*; Aros, *Athalia rosae*; Bmor, *Bombyx mori*; Cppal, *Culex pipiens pallens*; Dmel, *Drosophila melanogaster*; Gmor, *Glossina morsitans*; Llon, *Lutzomyia longipalpis*; Nvit, *Nasonia vitripennis*; Sfru, *Spodoptera frugiperda*; and Tcas, *Tribolium castaneum*. The accession numbers of the sequences used are shown in Appendix A.

**Figure 2 insects-14-00941-f002:**
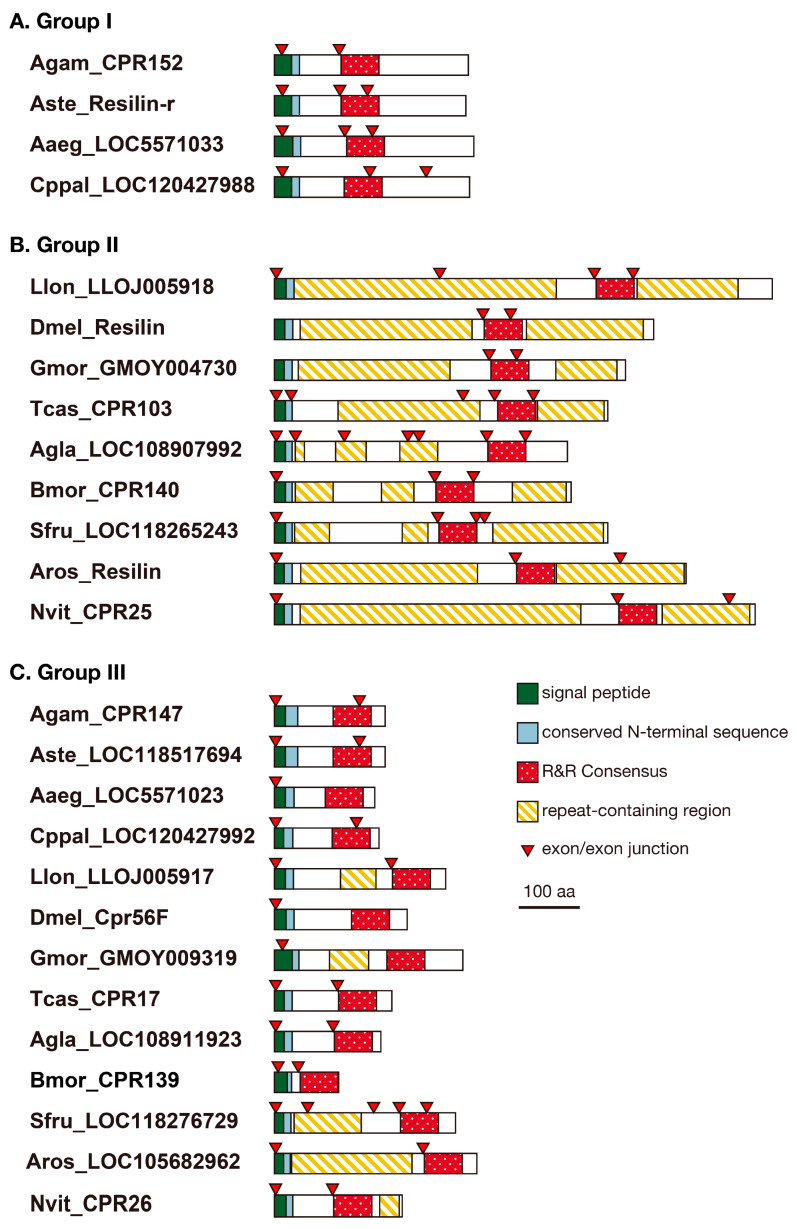
Primary structures of resilin homologous proteins. Schematic representations of the primary structures of resilin homologous proteins studied in this study are shown. (**A**) Group I; (**B**) Group II; (**C**) Group III; (**D**) Group IV proteins. The signal peptide, conserved N-terminal sequence, R&R Consensus, repeat-containing region, and exon/exon junction are indicated.

**Figure 3 insects-14-00941-f003:**
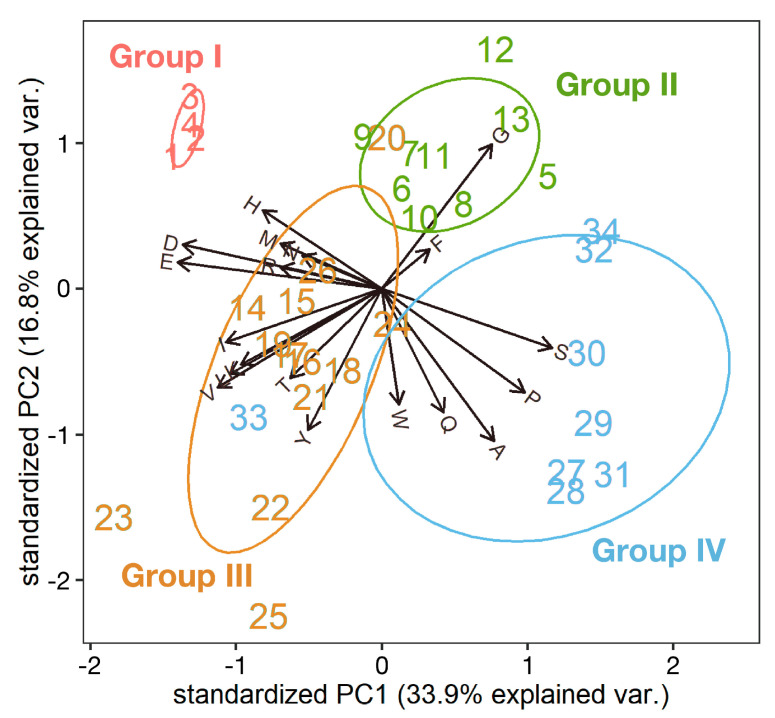
Principal component analysis (PCA) plot of resilin homologous proteins for their amino acid compositions. The PCA used the proportion of amino acid residues in each protein after removal of the predicted signal peptide. Arrows indicate the importance and correlation of each amino acid used as a variable on the two significant axes (PC1 and PC2). Red, green, orange, and blue letters indicate Group I, II, III, and IV proteins, respectively. Group I: 1, Agam_CPR152; 2, Aste_Resilin-r; 3, Aaeg_LOC5571033; 4, Cppal_LOC120427988. Group II: 5, Llon_LLOJ005918; 6, Dmel_Resilin; 7, Gmor_GMOY004730; 8, Tcas_CPR103; 9, Agla_LOC108907992; 10, Bmor_CPR140; 11, Sfru_LOC118265243; 12, Aros_Resilin; 13, Nvit_CPR25. Group III, 14, Agam_CPR147; 15, Aste_LOC118517694; 16, Aaeg_LOC5571023; 17, Cppal_LOC120427992; 18, Llon_LLOJ005917; 19, Dmel_Cpr56F; 20, Gmor_GMOY009319; 21, Tcas_CPR17; 22, Agla_LOC108911923; 23, Bmor_CPR139; 24, Sfru_LOC118276729; 25, Aros_LOC105682962; 26, Nvit_CPR26. Group IV: 27, Agam_AGAP002367; 28, Aste_LOC118502851; 29, Aaeg_LOC110681494; 30, Cppal_LOC120422587; 31, Dmel_Muc91C; 32, Tcas_LOC664382; 33, Bmor_LOC101740769; 34, Nvit_LOC100121527.

**Figure 4 insects-14-00941-f004:**
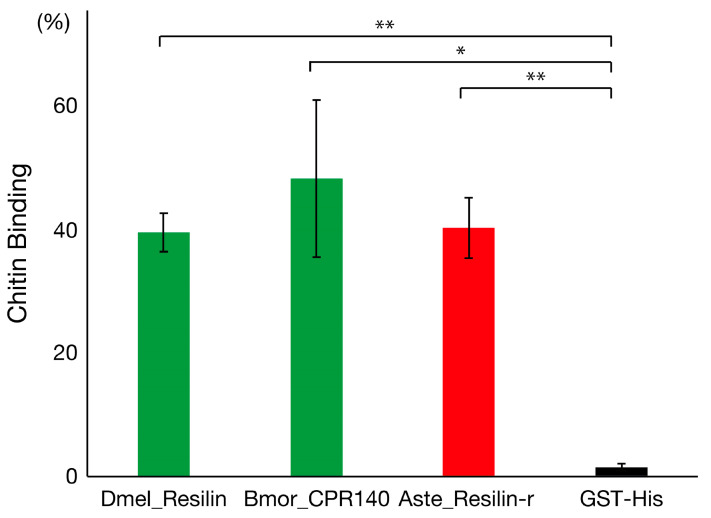
Chitin-binding activity of the R&R Consensus region of resilin and resilin-r. The chitin-binding activity of recombinant proteins of the R&R Consensus region of resilin from *D. melanogaster* (Dmel_Resilin) and *B. mori* (Bmor_CPR140), and resilin-r from *Anopheles stephensi* (Aste_Resilin-r) by chitin-affinity chromatography. The binding activity shows the ratio of the protein amount bound to chitin to the input protein determined by densitometry after SDS-PAGE of each fraction. *GST-His* represents the negative control protein without the R&R Consensus. The error bars indicate the S.E. of the mean, and asterisks indicate statistically significant differences (** *p* < 0.01; * *p* < 0.05) determined by *t* test using three experiments.

**Figure 5 insects-14-00941-f005:**
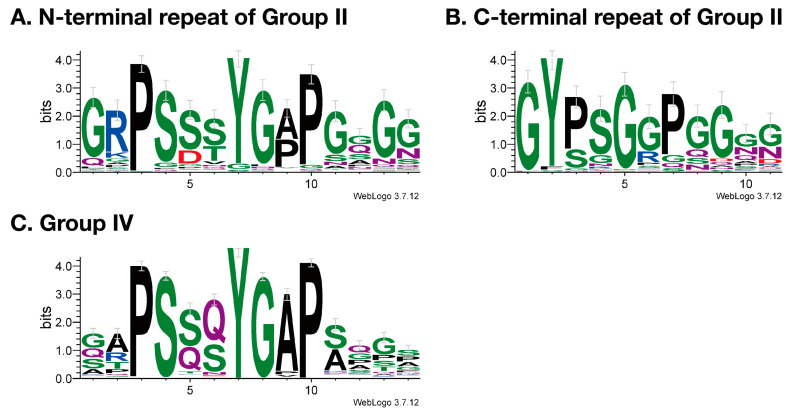
Core sequences of the repetitive region. WebLogos were constructed using all repetitive amino acid sequences of 14 residues in N-terminal (**A**) and 11 residues in C-terminal (**B**) regions to the R&R Consensus of Group II proteins and 14 residues in Group IV proteins (**C**).

## Data Availability

The sequence data used in this study are available in the GenBank/EMBL/DDBJ databases or VectorBase (https://vectorbase.org/vectorbase/app) (accessed from 23 August 2022 through 24 November 2023). The accession numbers and IDs are shown in Appendix A.

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
