# Peer review of "Mosquitoes Possess Specialized Cuticular Proteins That Are Evolutionarily Related to the Elastic Protein Resilin"

_insects, 2023, doi:10.3390/insects14120941_

Round 1

Reviewer 1 Report

Comments and Suggestions for Authors

In the manuscript, the authors describe a new family of proteins (resilin-r) from mosquitoes that have sequences similar to chitin binding sequences of insect resilin. They observed domains or motifs in the protein sequences of the resilin-r and their related proteins including resilin itself. With the analysis of molecular phylogenetic tree, they concluded that resilin-r is a kind of resilin-like proteins that were derived from resilin gene of mosquito. They also performed protein analysis for chitin-binding capacity with recombinant proteins, and showed that the chitin-binding motif of resilin-r is functional. The manuscript was written well, but I felt that chitin biding assay is the only and novel result obtained through experiments (though they determined sequences of cDNAs cloned by themselves). Additional data through experiments (at least expression pattern, opitonaly KD?) or more detailed bioinformatic analyses seem to be desired. My opinion is major revision requiring more data and descriptions

I think that there are other major points that they should consider before publication. 

1. the authors used only a limited numbers of sequences for the phylogenetic analysis. This is because (I guess) they wanted to use only the proteins that they can manipulate in their laboratory, but they used also other sequences of mosquitos without experimental confirmation. Therefore, I felt that they can extend the number of species used in phylogenetic analysis to make the result more precise or conclusive?  Why did the authors use several species only in mosquito? 

2. related to 1., with more sequences from multiple insect orders, it will be possible to show the mosquito proteins (resilin-r) are the characteristics found only in mosquitos or the resilin-r-like molecules are found in other insects (especially dipterans). From the manuscript, it is unclear whether resilin-r is the mosquito proteins only found in mosquito or present in other insects ?

3. in the manuscript, the authors describe amino acid compositions, but not at all about the details on the repeats responsible for the elasticity. Presence or absence of the repeats affects amino acid compositions of the proteins that they discuss. This manuscript describes proteins related to resillin, but the explanation for resilin itself seems insufficient. the important thing may be absence of explanations for the repeats. In the first report for putative cDNA of drosophila resilin, Ardel and Andersen explained the details of the repeat sequences, and the following molecular studies have been focusing on the function of the repeats in forming elastic materials. In fig.3 and .5 the authors mention the characteristics in the sequence of the repeats, but it is not described what kinds of sequences are common in the repeats of resilin molecules. In the manuscript, they say “proteins that are annotated as pro-resilin” or like that, but the names of these proteins were maybe automatically assigned. Without explanations of the repeats, it seems to be difficult for readers to understand that the group4 proteins have really the repeats commonly shared by resilins or resilin-like proteins. One more thing, are there any group4 proteins (with only repeats) in any other orders of insects?  The main object of this manuscript is chitin binding domain, but it is good to explain more about other structures, especially the repeats unique to resilin.  

4. chitin-biding assay described here is a kind of static-state measurement, because the binding was not monitored on time, but a comparison between protein bound and unbound after application to the column.The unbound fractions are possibly containing mis-folded proteins and thus un-functional. If the authors discuss on the strength of the affinity to chitin, is it possible to measure by adding a series of urea concentration to monitor how each recombinant protein remain to bind to chitin strongly? Or it seems to be able just to say that at least each recombinant protein has affinity to chitin (though I think this is enough). 

line 83-5: is it kind to show alignment of the sequences used in the analysis?

line169: I am not sure BmorCPR139 can be grouped in GIII from the branching pattern of the tree?

until218: explanations for structural characters are maybe good, but the meanings in the analyses of dot plots are not explained properly. or is it required to show the results, if the authors do not explain well?

line 321: is ancient required?, because before divergence of the mosquito ancestor, it seems that resilin gene had been functionally completed, having the common structure with chitin binding motif and the N- and/or C-terminal repeats even in ancestral insects of flies and mosquitos (maybe)?  

until218: explanations for structural characters are maybe good, but the meanings in the analyses of dot plots are not explained properly. or is it required to show the results, if the authors do not explain well?

line 321: is ancient required?, because before divergence of the mosquito ancestor, it seems that resilin gene had been functionally completed, having the common structure with chitin binding motif and the N- and/or C-terminal repeats even in ancestral insects of flies and mosquitos (maybe)?  

Comments on the Quality of English Language

good

Author Response

In the manuscript, the authors describe a new family of proteins (resilin-r) from mosquitoes that have sequences similar to chitin binding sequences of insect resilin. They observed domains or motifs in the protein sequences of the resilin-r and their related proteins including resilin itself. With the analysis of molecular phylogenetic tree, they concluded that resilin-r is a kind of resilin-like proteins that were derived from resilin gene of mosquito. They also performed protein analysis for chitin-binding capacity with recombinant proteins, and showed that the chitin-binding motif of resilin-r is functional. The manuscript was written well, but I felt that chitin biding assay is the only and novel result obtained through experiments (though they determined sequences of cDNAs cloned by themselves). Additional data through experiments (at least expression pattern, opitonaly KD?) or more detailed bioinformatic analyses seem to be desired. My opinion is major revision requiring more data and descriptions

Response:

Thank you very much for the valuable comments. We understand that the current study provides few experimental data. The main purpose of this study is to reveal the ubiquity of resilin-related in mosquitoes and other insects, and to clarify its relationship to other resilin homologous proteins. The functional study is the future project. However, we agree that this manuscript needs more information. We included more resilin homologous sequences from other insect species into the phylogenetic analysis. We also analyzed more carefully the repetitive sequences in Group II and IV proteins according to your valuable suggestion and found clearer relationship between Group II and IV proteins.

I think that there are other major points that they should consider before publication. 

1. the authors used only a limited numbers of sequences for the phylogenetic analysis. This is because (I guess) they wanted to use only the proteins that they can manipulate in their laboratory, but they used also other sequences of mosquitos without experimental confirmation. Therefore, I felt that they can extend the number of species used in phylogenetic analysis to make the result more precise or conclusive?  Why did the authors use several species only in mosquito? 

Response:

Thank you very much for the valuable suggestion. We included more resilin homologous sequences from other insect species into the phylogenetic analysis and revised Figure 1 and text according to the new results.

2. related to 1., with more sequences from multiple insect orders, it will be possible to show the mosquito proteins (resilin-r) are the characteristics found only in mosquitos or the resilin-r-like molecules are found in other insects (especially dipterans). From the manuscript, it is unclear whether resilin-r is the mosquito proteins only found in mosquito or present in other insects ?

Response:

Thank you very much for your valuable comment. We did not find resilin-r out of mosquitoes. Resilin-r is likely mosquito specific. We clearly stated about it at line 238 and 241.

3. in the manuscript, the authors describe amino acid compositions, but not at all about the details on the repeats responsible for the elasticity. Presence or absence of the repeats affects amino acid compositions of the proteins that they discuss. This manuscript describes proteins related to resillin, but the explanation for resilin itself seems insufficient. the important thing may be absence of explanations for the repeats. In the first report for putative cDNA of drosophila resilin, Ardel and Andersen explained the details of the repeat sequences, and the following molecular studies have been focusing on the function of the repeats in forming elastic materials. In fig.3 and .5 the authors mention the characteristics in the sequence of the repeats, but it is not described what kinds of sequences are common in the repeats of resilin molecules. In the manuscript, they say “proteins that are annotated as pro-resilin” or like that, but the names of these proteins were maybe automatically assigned. Without explanations of the repeats, it seems to be difficult for readers to understand that the group4 proteins have really the repeats commonly shared by resilins or resilin-like proteins. One more thing, are there any group4 proteins (with only repeats) in any other orders of insects?  The main object of this manuscript is chitin binding domain, but it is good to explain more about other structures, especially the repeats unique to resilin.  

Response:

Thank you very much for the valuable suggestions. We further analyzed the repetitive sequences in Group II and IV proteins. We realized that Group IV proteins exist commonly in holometabolous insects, and they shared structural features that is also shared with Group II proteins. We added description about it in the discussion section.

4. chitin-biding assay described here is a kind of static-state measurement, because the binding was not monitored on time, but a comparison between protein bound and unbound after application to the column.The unbound fractions are possibly containing mis-folded proteins and thus un-functional. If the authors discuss on the strength of the affinity to chitin, is it possible to measure by adding a series of urea concentration to monitor how each recombinant protein remain to bind to chitin strongly? Or it seems to be able just to say that at least each recombinant protein has affinity to chitin (though I think this is enough). 

Response:

Thank you very much for the valuable suggestions. We canceled to express “comparable chitin-binding activity” and changed to “showed chitin-binding activity as same as that of resilin” (line 273).

line 83-5: is it kind to show alignment of the sequences used in the analysis?

Response:

We included the alignment used for the phylogenetic analysis as Figure S1.

line169: I am not sure BmorCPR139 can be grouped in GIII from the branching pattern of the tree?

Response:

Using more sequences replaced BmorCPR139 into an ingroup. We revised Figure 1.

until218: explanations for structural characters are maybe good, but the meanings in the analyses of dot plots are not explained properly. or is it required to show the results, if the authors do not explain well?

Response:

Thank you very much for the kind suggestion. We canceled to use dot plots. We use schematic representation to show the primary structures as new Figure 2.

line 321: is ancient required?, because before divergence of the mosquito ancestor, it seems that resilin gene had been functionally completed, having the common structure with chitin binding motif and the N- and/or C-terminal repeats even in ancestral insects of flies and mosquitos (maybe)?  

Response:

Thank you for the valuable suggestion. We revised the text at line 350.

Reviewer 2 Report

Comments and Suggestions for Authors

The manuscript from Ohkubo et al investigated the relationship between resilin and resilin-r in some species of holometabolous insects. Most of the conclusions drawn by the authors are only supported by in silico analysis. Overall the manuscript contains some novelty and resulted in the in silico identification of an additional resilin in mosquitoes that was named resilin-like by the authors.

Specific points.

1.   1. Please include in the supplementary figures images of the gels that are mentioned in lines 138-140 (Materials and Methods).

2.    2. Regarding the phylogenetical analysis

It is unclear why the taxon sampling was restricted to the investigated species, namely Anopheles gambiae, Anopheles stephensi, Aedes aegypti, Culex pipiens pallens, D. melanogaster, Tribolium castaneum, Bombyx mori, and Athalia rosae. For instance, why Culex pipiens pallens and not Culex quinquefasciatus?

It is important to justify the selection, especially because the set of sequences were used in phylogenetical analyses. Moreover, the phylogenetical distances between B. mori, A. rosae and T. castaneum and the remaining Diptera are very large. Are the authors sure that it did not cause distortions in the analyses? Overall the bootstrap values are not very good. Please comment.

3.    3. Regarding Supplementary Figure 1.

The authors should consider including this figure as one of the main figures in the text. It is informative and helps the reader.

4.    4. PCA analysis

Why group III was excluded from the PCA? Figure 3 has to be improved, at present it is a very confusing figure. Please revise the text (lines 234 to 239). The conclusions that are presented cannot be drawn solely from the PCA analysis.

5.    5. Figure 4. Please clarify what is the negative control mentioned in the legend of figure 4 (lines 265-266). Also please show as a supplementary figure the SDS-PAGE representative results that originated the graph in figure 4.

6.    6. Please revise the entire discussion. Ex. “somehow convincing” (line 298). Increasing the taxon sampling would improve the manuscript. Why proteins from group II “seem to function” as resilin (lines 300-301)?. Please further discuss what would be the role of Group I resilins. With the exception of D. melanogaster, all investigated Diptera species are from the Culicidae family. Are group I and group IV resilins also present in other families of the Culicomorpha infraorder? That would change the evolutionary hypothesis. Also please make clear in the discussion the actual contribution of the study to the field.

Author Response

The manuscript from Ohkubo et al investigated the relationship between resilin and resilin-r in some species of holometabolous insects. Most of the conclusions drawn by the authors are only supported by in silico analysis. Overall the manuscript contains some novelty and resulted in the in silico identification of an additional resilin in mosquitoes that was named resilin-like by the authors.

Specific points.

  1.  1. Please include in the supplementary figures images of the gels that are mentioned in lines 138-140 (Materials and Methods).

Response:

Thank you very much for the suggestion. We added gel data as new Figure S2.

  1. 2. Regarding the phylogenetical analysis

It is unclear why the taxon sampling was restricted to the investigated species, namely Anopheles gambiaeAnopheles stephensiAedes aegyptiCulex pipiens pallensD. melanogasterTribolium castaneumBombyx mori, and Athalia rosae. For instance, why Culex pipiens pallens and not Culex quinquefasciatus?

It is important to justify the selection, especially because the set of sequences were used in phylogenetical analyses. Moreover, the phylogenetical distances between B. mori, A. rosae and T. castaneum and the remaining Diptera are very large. Are the authors sure that it did not cause distortions in the analyses? Overall the bootstrap values are not very good. Please comment. 

Response:

Thank you very much for the valuable suggestions. We included more resilin homologous sequences from other insect species into the phylogenetic analysis and revised the manuscript accordingly. Culex quinquefasciatus is certainly studied well, but Culex pipiens is also studied well. We believe that using C. pipiens provide essentially the same results as using C. quinquefasciatus. We have realized that the bootstrap value was not so good, too. At least it can be said that the resilin homologous proteins used in this study, which are most and secondary most similar to the Drosophila resilin, were isolated from the outgroup consisting of the tertiary most similar proteins to the resilin. Other structural features, the conserved N-terminal sequences and the presence of repetitive sequences, may support grouping the Group II and III, although the phylogenetic divergence between them is not clear. We stated clearly about it at lines 181 and 391.

  1. 3. Regarding Supplementary Figure 1. 

The authors should consider including this figure as one of the main figures in the text. It is informative and helps the reader. 

Response:

Thank you very much for the valuable suggestion. We adopted old supplementary Figure S1 as new Figure 2.

  1. 4. PCA analysis

Why group III was excluded from the PCA? Figure 3 has to be improved, at present it is a very confusing figure. Please revise the text (lines 234 to 239). The conclusions that are presented cannot be drawn solely from the PCA analysis. 

Response:

Thank you very much for the valuable suggestion. We performed PCA again with Group III proteins. The details about amino acid composition are shown in supplementary Table S4. Therefore, we added reference to Table S4.

  1. 5. Figure 4. Please clarify what is the negative control mentioned in the legend of figure 4 (lines 265-266). Also please show as a supplementary figure the SDS-PAGE representative results that originated the graph in figure 4.

Response:

Thank you very much for the valuable suggestion. We added the explanations for the negative control and for other analyzed proteins at lines 269-272. We added gel data as new Figure S2.

  1. 6. Please revise the entire discussion. Ex. “somehow convincing” (line 298). Increasing the taxon sampling would improve the manuscript. Why proteins from group II “seem to function” as resilin (lines 300-301)?. Please further discuss what would be the role of Group I resilins. With the exception of D. melanogaster, all investigated Diptera species are from the Culicidae family. Are group I and group IV resilins also present in other families of the Culicomorpha infraorder? That would change the evolutionary hypothesis. Also please make clear in the discussion the actual contribution of the study to the field.

Response:

Thank you very much for the valuable suggestion. We revised the entire discussion and added the Conclusion section. We tried to use sequences from other Culicomorpha. We found that Culicoides sonorensis genome is available, but annotation is not good. We could not find resilin homologous transcripts from this species. We found resilin homologs from Lutzomyia longipalpis, another species of Nematocera suborder. These sequences were included into the analyses, and we found that this species does not have mosquito-specific resilin-related but have a canonical resilin.

Round 2

Reviewer 1 Report

Comments and Suggestions for Authors

Dear the authors,

The manuscript was improved very much by considering the comments, especially they describe the details of repetitive sequences in resilin and resilin-like proteins. Also the phylogenetic tree was improved by adding sequences (I felt that it will be better to use more sequences?). The similarity between the N-terminal repeats of the group 2 and the repeats of group 4 is important to discuss the evolutional events to produce such new genes.

In my review, I recommended to add experimental evidence Or sequence analysis data, but the authors nicely provided the improved (and new) data of the sequence analyses.

My recommendation is accept (with corrections of errors or mistakes) (sorry I did not point them out, because I could not find, but there may be. Please check it.) 

Author Response

Dear the authors,

The manuscript was improved very much by considering the comments, especially they describe the details of repetitive sequences in resilin and resilin-like proteins. Also the phylogenetic tree was improved by adding sequences (I felt that it will be better to use more sequences?). The similarity between the N-terminal repeats of the group 2 and the repeats of group 4 is important to discuss the evolutional events to produce such new genes.

In my review, I recommended to add experimental evidence Or sequence analysis data, but the authors nicely provided the improved (and new) data of the sequence analyses.

My recommendation is accept (with corrections of errors or mistakes) (sorry I did not point them out, because I could not find, but there may be. Please check it.) 

Response:

Thank you very much for your kind comments. The manuscript was highly improved because of your valuable comments and suggestions. We deeply appreciate it.

Reviewer 2 Report

Comments and Suggestions for Authors

The authors reviewed the entire manuscript and addressed all the queries that have been made. The manuscript is clearer and the figures have improved. No further modifications are suggested.

Author Response

The authors reviewed the entire manuscript and addressed all the queries that have been made. The manuscript is clearer and the figures have improved. No further modifications are suggested.

Response:

Thank you very much for your kind comments. The manuscript was highly improved because of your valuable comments and suggestions. We deeply appreciate it.